# Prediction of Plant Phenological Shift under Climate Change in South Korea

**Ha Kyung Lee, So Jeong Lee, Min Kyung Kim and Sang Don Lee ***

Department of Environmental Science and Engineering, College of Engineering, Ewha Womans University, Seoul 03760, Korea; hklee831@gmail.com (H.K.L.); lsojeong@naver.com (S.J.L.); enviecol@ewha.ac.kr (M.K.K.)
* Correspondence: lsd@ewha.ac.kr; Tel.: +82-232-773-545; Fax: +82-232-773-275

**Abstract:** Information on the phenological shift of plants can be used to detect climate change and predict changes in the ecosystem. In this study, the changes in first flowering dates (FFDs) of the plum tree (*Prunus mume*), Korean forsythia (*Forsythia koreana*), Korean rosebay (*Rhododendron mucronulatum*), cherry tree (*Prunus yedoensis*), and peach tree (*Prunus persica*) in Korea during 1920–2019 were investigated. In addition, the changes in the climatic factors (temperature and precipitation) and their relationship with the FFDs were analyzed. The changes in the temperature and precipitation during the January–February–March period and the phenological shifts of all research species during 1920–2019 indicate that warm and dry spring weather advances the FFDs. Moreover, the temperature has a greater impact on this phenological shift than precipitation. Earlier flowering species are more likely to advance their FFDs than later flowering species. Hence, the temporal asynchrony among plant species will become worse with climate change. In addition, the FFDs in 2100 were predicted based on representative concentration pathway (RCP) scenarios. The difference between the predicted FFDs of the RCP 4.5 and RCP 6.0 for 2100 was significant; the effectiveness of greenhouse gas policies will presumably determine the degree of the plant phenological shift in the future. Furthermore, we presented the predicted FFDs for 2100.

**Keywords:** plant phenology; climate change; first flowering date; RCP scenario; temperature; asynchrony

---

## 1. Introduction

Seasonally repeated phenological events (e.g., flowering, leaf falling, first appearance of migratory birds, and breeding of vertebrates) can be studied to determine the effects of climate change [1–3]. As environmental conditions have been changed due to climate change, diverse of taxonomic groups have shown a shift in phenological events such as earlier onset of vegetation activity in spring, extension in the length of the growing season [4], modified distribution of insects and changes to insect migration routes [5,6], and mistimed avian reproduction [7]. However, different taxonomic groups have shown dissimilar rates of phenological change and plants are the one that have shown advanced spring phenology in a consistent manner, whereas other taxonomic groups have shown more divergent responses and have shifted their phenology less. Plants are highly sensitive to climatic changes because they are sessile organisms [8–10]. In addition, they are producers at the bottom of the trophic level; their phenological change can influence the upper taxonomic groups such as pollinators and predators. Consequently, the changes in plant phenology can cause mismatches between interacting species, resulting in a fundamental disruption of the structure of ecological interactions [11–17]. Therefore, the prediction of plant phenological shift can provide information on how ecological interactions and structure would transfigure in the future.

The majority of previous phenology studies have described phenology shifts in terms of thermal conditions. Temperature has been proven to be the most significant factor in plant phenology.

It shows the highest relevance for plant phenology than other climatic factors such as sunshine, frost, and snowmelt [4,18–22]. This high relevance can be explained by the relation between temperature and underlying biochemical processes of ectotherms such as reaction kinetics, enzyme inactivation, and hormonal regulation [23–27]. Precipitation, which describes the moisture content of soil, also could be a factor that can affect plant phenology, especially in a dry environment [28–31]. Because temperature and precipitation are expected to vary greatly owing to climate change, identifying the relationship between temperature and precipitation and plant phenology is crucial for identifying plant phenological shifts caused by climate change.

According to the fifth assessment report of the Intergovernmental Panel on Climate Change (IPCC) based on representative concentration pathway (RCP) scenarios, the average global temperature will increase by 3.7 °C, and the average temperature in Korea will increase by 6 °C by the end of the 21st century. As the temperature has such an important effect on plants, it is necessary to study how much of this level of climate change would affect plant phenology. However, phenological shift has temporal and spatial variations even within the same species [22,32,33]. These variations indicate that phenology investigation should be done in fine scales for individual species. The Korea Meteorological Administration (KMA) has recorded the first flowering date (FFD) of several designated phenological observation species over 72 locations since 1920. It also provides detailed climate change scenarios for the Korean Peninsula up to 2100 with a resolution of 1 km based on RCP scenarios. This information enabled us to investigate the plant phenological changes nationwide in detail.

In this study, the changes in the FFDs over the past 100 years of the six early spring flowering plants (the plum tree (*Prunus mume*), Korean forsythia (*Forsythia koreana*), Korean rosebay (*Rhododendron mucronulatum*), cherry tree (*Prunus yedoensis*), peach tree (*Prunus persica*), and pear tree (*Pyrus serotina*)) over 72 location in Korea were investigated. Subsequently, the most influential climatic factor for plant species in Korea was determined between temperature and precipitation and applied to the plant phenological model to predict FFDs in Korea in 2100. This model we developed allowed us to infer the future trend of plant phenology in Korea across the range of each species by using long time series of phenology data from numerous locations distributed throughout the country. We were also able to see the future FFDs at locations for which we do not have historic records by creating a continuous surface model. This study seeks to forecast how much climate change would induce phenological changes.

## 2. Materials and Methods

### 2.1. Phenological Data

Since 1920, the KMA has been recording phenological data for several designated plant and animal species from more than 72 locations in Korea (Figure 1, Table 1; data available from KMA, https://data.kma.go.kr/data/seasonObs/seasonObsDataList.do?pgmNo=648). For this study, six of the designated plant species which have their FFDs in spring were chosen (the plum tree, Korean forsythia, Korean rosebay, cherry tree, peach tree, and pear tree; Table 2). The FFD data from 1920 to 2019 were converted into Julian dates for the statistical analysis. The observed designated plant species are kept as natural as possible at the same place every year, with minimal maintenance for preventing diseases. This data is much more credible because it has been systematically collected by a government body at the same locality. Yet, some observation sites have data gaps during the 1950s due to the Korean War. Missing values (data neither recorded nor monitored) were excluded from all statistical analyses. The FFD refers to when a bud sprouts, but for a plant which has many flowers in one branch, it refers to the time when more than three flowers bloom on any branch. We first measured long-term phenological shifts by the slopes of linear models for FFD against year. We also conducted a regression analysis of the FFDs of each plant species of the 72 observation sites with respect to the year to examine the variations in the phenological trend of each species.

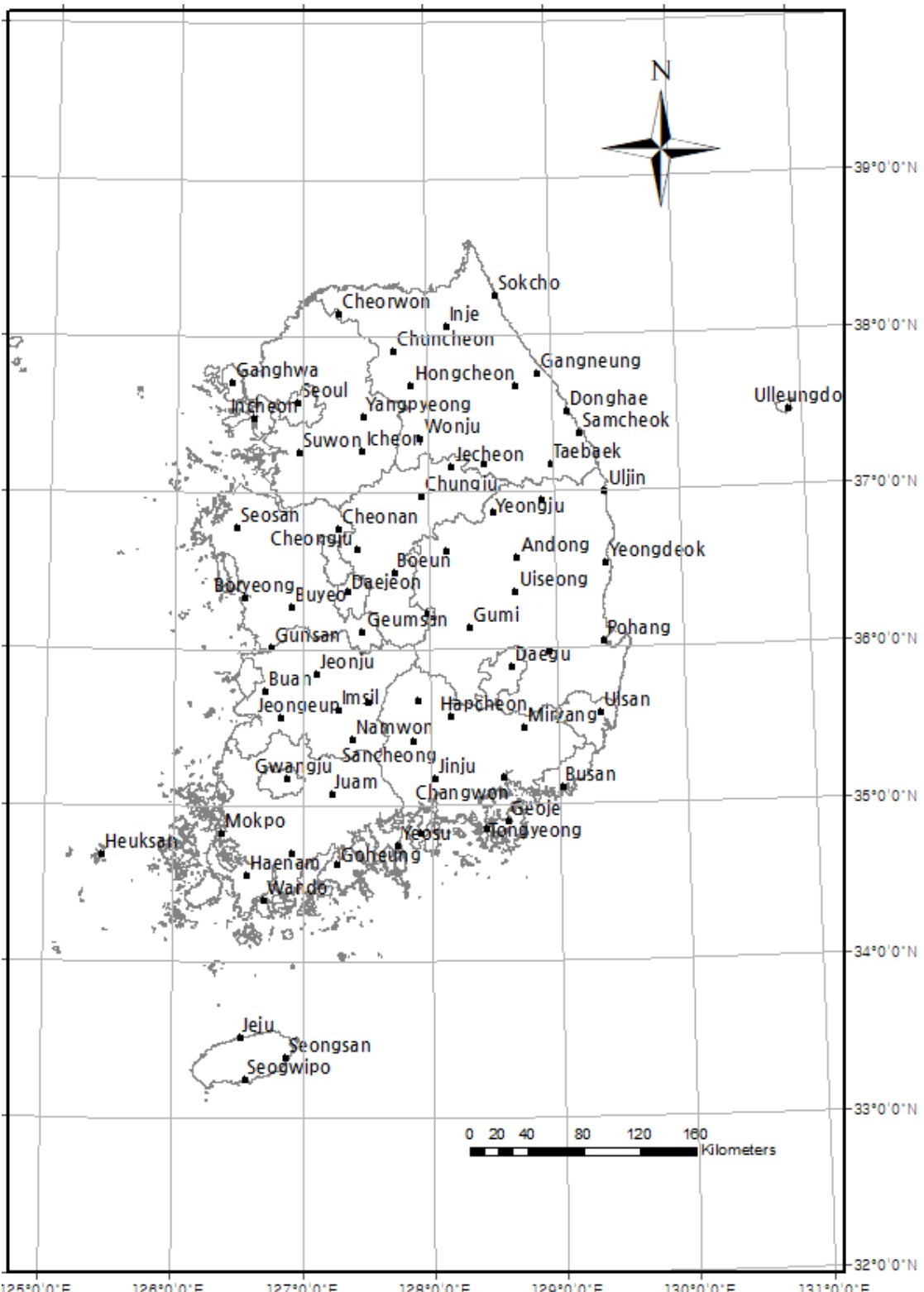

**Figure 1.** Location of 72 phenological and meteorological observation sites during 1920–2019 in Korea.

**Table 1.** Location of 72 phenological and meteorological observation sites during 1920–2019 in Korea.

| No | Station | Latitude (°N) | Longitude (°E) | Altitude (m) |
|---|---|---|---|---|
| 105 | Gangneung | 37.75 | 128.89 | 27 |
| 201 | Ganghwa | 37.71 | 126.45 | 48 |
| 294 | Geoje | 34.89 | 128.6 | 45 |
| 284 | Geochang | 35.67 | 127.91 | 228 |
| 262 | Goheung | 34.62 | 127.28 | 52 |
| 156 | Gwangju | 35.17 | 126.89 | 72 |
| 279 | Gumi | 36.13 | 128.32 | 49 |
| 140 | Gunsan | 36.01 | 126.76 | 28 |
| 238 | Geumsan | 36.11 | 127.48 | 173 |
| 247 | Namwon | 35.42 | 127.4 | 133 |
| 295 | Namhae | 34.82 | 127.93 | 46 |
| 100 | Daegwallyeong | 37.68 | 128.72 | 772 |
| 143 | Daegu | 35.88 | 128.65 | 54 |
| 133 | Daejeon | 36.37 | 127.37 | 70 |
| 106 | Donghae | 37.51 | 129.12 | 40 |
| 165 | Mokpo | 34.82 | 126.38 | 45 |
| 273 | Mungyeong | 36.63 | 128.15 | 173 |
| 288 | Miryang | 35.49 | 128.74 | 11 |
| 235 | Boryeong | 36.33 | 126.56 | 10 |
| 226 | Boeun | 36.49 | 127.73 | 171 |
| 271 | Bonghwa | 36.94 | 128.91 | 325 |
| 159 | Busan | 35.1 | 129.03 | 70 |
| 243 | Buan | 35.73 | 126.72 | 12 |
| 236 | Buyeo | 36.27 | 126.92 | 13 |
| 289 | Sancheong | 35.41 | 127.88 | 138 |
| 214 | Samcheok | 37.37 | 129.22 | 4 |
| 189 | Seogwipo | 33.25 | 126.57 | 52 |
| 129 | Seosan | 36.78 | 126.49 | 25 |
| 108 | Seoul | 37.57 | 126.97 | 86 |
| 188 | Seongsan | 33.39 | 126.88 | 20 |
| 265 | Seongsanpo | 33.39 | 126.88 | 19 |
| 90 | Sokcho | 38.25 | 128.56 | 18 |
| 119 | Suwon | 37.26 | 126.98 | 40 |
| 136 | Andong | 36.57 | 128.71 | 141 |
| 202 | Yangpyeong | 37.49 | 127.49 | 47 |
| 168 | Yeosu | 34.74 | 127.74 | 65 |
| 277 | Yeongdeok | 36.53 | 129.41 | 41 |
| 121 | Yeongwol | 37.18 | 128.46 | 241 |
| 272 | Yeongju | 36.87 | 128.52 | 211 |
| 281 | Yeongcheon | 35.98 | 128.95 | 96 |
| 170 | Wando | 34.4 | 126.7 | 35 |
| 115 | Ulleungdo | 37.48 | 130.9 | 222 |
| 152 | Ulsan | 35.58 | 129.33 | 81 |
| 130 | Uljin | 36.99 | 129.41 | 49 |
| 114 | Wonju | 37.34 | 127.95 | 150 |
| 278 | Uiseong | 36.36 | 128.69 | 81 |
| 203 | Icheon | 37.26 | 127.48 | 80 |
| 211 | Inje | 38.06 | 128.17 | 202 |
| 112 | Incheon | 37.48 | 126.62 | 69 |
| 244 | Imsil | 35.61 | 127.29 | 247 |
| 248 | Jangsu | 35.66 | 127.52 | 406 |
| 260 | Jangheung | 34.69 | 126.92 | 45 |
| 146 | Jeonju | 35.84 | 127.12 | 60 |
| 245 | Jeongeup | 35.56 | 126.84 | 69 |
| 184 | Jeju | 33.51 | 126.53 | 21 |

**Table 1.** *Cont.*

| No | Station | Latitude (°N) | Longitude (°E) | Altitude (m) |
|---|---|---|---|---|
| 221 | Jecheon | 37.16 | 128.19 | 265 |
| 256 | Juam | 35.08 | 127.24 | 75 |
| 192 | Jinju | 35.16 | 128.04 | 29 |
| 155 | Changwon | 35.17 | 128.57 | 38 |
| 232 | Cheonan | 36.76 | 127.29 | 85 |
| 95 | Cheorwon | 38.15 | 127.3 | 155 |
| 131 | Cheongju | 36.64 | 127.44 | 59 |
| 135 | Chupungnyeong | 36.22 | 127.99 | 245 |
| 101 | Chuncheon | 37.9 | 127.74 | 76 |
| 127 | Chungju | 36.97 | 127.95 | 115 |
| 216 | Taebaek | 37.17 | 128.99 | 714 |
| 162 | Tongyeong | 34.85 | 128.44 | 32 |
| 138 | Pohang | 36.03 | 129.38 | 4 |
| 285 | Hapcheon | 35.57 | 128.17 | 32 |
| 261 | Haenam | 34.55 | 126.57 | 16 |
| 212 | Hongcheon | 37.68 | 127.88 | 140 |
| 169 | Heuksan | 34.69 | 125.45 | 76 |

**Table 2.** List of species used for phenological analysis in this study (FFD: first flowering date).

| Species | | FFD Observation Period |
|---|---|---|
| **Common Name** | **Scientific Name** | |
| Plum tree | *Prunus mume* | late January~mid-May |
| Korean forsythia | *Forsythia koreana* | late February~mid-May |
| Korean rosebay | *Rhododendron mucronulatum* | early March~late April |
| Cherry tree | *Prunus yedoensis* | early March~late April |
| Peach tree | *Prunus persica* | Mid-March~early May |
| Pear tree | *Pyrus serotina* | late March~mid-May |

*2.2. Climatic Data*

In this study, the temperature and precipitation were considered as the most influential factors for plant phenology. The temperature and precipitation data from 1920 to 2019 originate from the plant phenology observation site of the KMA (Figure 1). The KMA has recorded the climatic data by ASOS (automated synoptic observing system). Temperature is recorded every hour and averaged to daily and monthly values in order. Precipitation is also recorded every hour and summed up to give daily and monthly values. Statistical values were calculated when the amount of data was more than 80%. Plant species with a first flowering date (FFD) in spring are known to be affected by the climate within the first to three months before their FFD, after winter dormancy [13,31,34]. In previous plant phenology studies conducted in Korea, it turns out that a combination of more than one month's climatic factor had a higher correlation with FFD [9,33,35]. We compared the determination coefficient ($R^2$) of two-month averaged climatic factors and three-month averaged climatic factors with the FFD in our analysis. For the final analysis, the climatic data for three months before the FFDs of each plant species were averaged to calculate the January–February–March average temperature (JFMT) and January–February–March average precipitation (JFMP), which showed the highest explanation ability across all species. To identify how much the FFDs and climatic factors have changed over the past century, the differences between the characteristics of 1920 and 2019 were calculated. In addition, regression analyses between FFDs and climatic factors were conducted to determine their relationships; R and SPSS (version 25.0) were used for all statistical analyses.

## 2.3. Plant Phenological Model

The KMA Climate Information Portal provides detailed climate change scenarios based on RCP scenarios for Korea (http://www.climate.go.kr/home/CCS/contents/33_1_areapoint.php). The RCP scenarios consists of four parts, including RCP 2.6 (instantaneous greenhouse gas reduction), RCP 4.5 (substantial achievement of greenhouse gas reduction policy), RCP 6.0 (fair achievement of greenhouse gas reduction policy), and RCP 8.5 (greenhouse gas emission as current trend). The FFD was analyzed as a function of the temperature and applied to all four of these scenarios to predict the FFD of each species in 2100 for the 72 locations (Figure 1). To create a continuous surface model that covers the entire country, future FFDs of unanalyzed locations were estimated by using kriging interpolation. Kriging interpolation is based on localized variables. Localized variables have positive autocorrelations and are greatly influenced by their distance from the sample points, which can be expressed as follows:

$$Z(x) = m(x) + \varepsilon'(x) + \varepsilon''$$

$Z(x) :$ *Localized variable at x location, which is FFD in this study.*

$m(x) :$ *Structural element indicating a constant tendency for Z at location x.*

$\varepsilon'(x) :$ *Local variation element of Z, which is changing with respect to spatial location.*

$\varepsilon'' :$ *An error occurring independently of spatial position*

Kriging interpolation method calculates the Z value of unexamined location based on the variogram which indicates the actual Z values of sample locations and the mean variance between the distances between these sample locations. The estimates of unexamined locations are calculated by weighting them to minimize variance. ArcGIS10.7 was used for the modeling.

## 3. Results and Discussion

### 3.1. Changes in Climatic Factors (Temperature and Precipitation) and Plant Phenology during 1920–2019

During 1920–2019, the average JFMT at the observation sites increased by 2.6 °C, and the average JFMP decreased by 18 mm, although it was not statistically significant, with high *p*-value (Figure 2). These changes fluctuated periodically. The averaged FFDs of the plant species at the observation sites also fluctuated with similar cycles to the climatic factors; nonetheless the overall changes were advanced by 10–53 days during 100 years (Table 3). These results agree with those of previous studies; warmer and drier spring weather advances the FFDs [10,36–39]. Every investigated plant species showed different shift rates. The shift rate of Korean rosebay (0.12 day/year), cherry tree (0.21 day/year), and peach tree (−0.11 day/year) showed almost the same value as a previous study which was conducted for 35 years (from 1973 to 2008) in Korea [33]. However, the plum tree, which is the earliest blooming species in this study, showed an exceptionally high shift rate of 0.53 day/year. The FFD of the plum tree has fluctuated greatly, showing low statistical validity with high *p*-value, and 2019 seems to be one of the low points of its fluctuation cycle. Thus, there is a possibility that the shift rate of the plum tree could have been overestimated. However, if the plum tree keeps advancing its FFD at the speed calculated in this study, it would be one of the extreme phenomena caused by climate change.

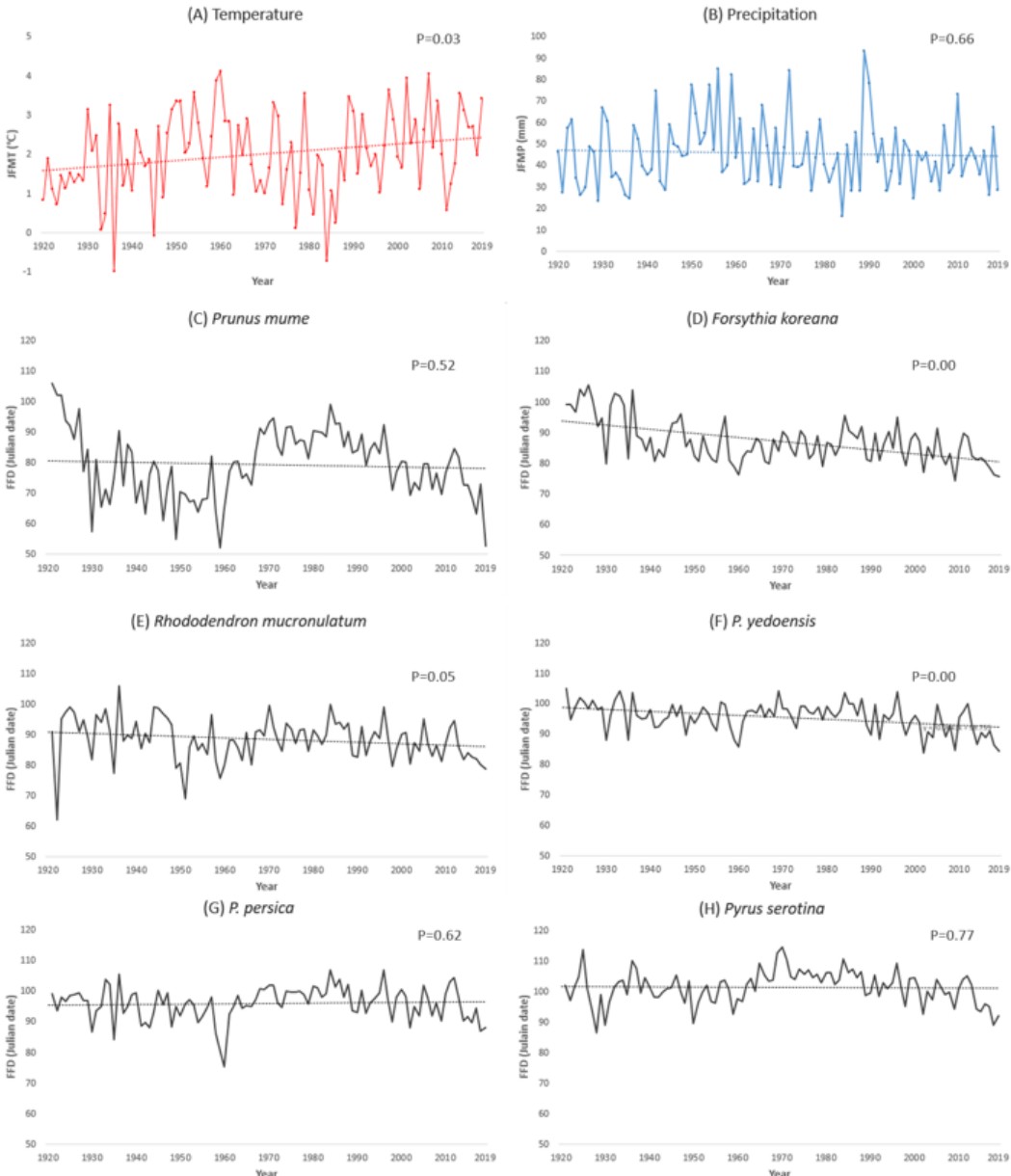

**Figure 2.** Change of January–February–March averaged (**A**) temperature, (**B**) precipitation, and (**C–H**) phenological change (FFD) of each plant species during 1920–2019 in Korea. *p*-values of regression analysis are also shown. (**A**) Temperature JFMT = 0.01 C/year +1.57, (**B**) precipitation JFMP = −0.03 mm/year +46.9 during 1920–2019 in Korea.

**Table 3.** FFDs in 2019 and total changes in FFD of each species during 1920–2019 in Korea.

| Species | FFD in 2019 (Julian Date) | Total Amount of Change (Day/100 Years) |
|---|---|---|
| *Prunus mume* | 53 | −53 |
| *Forsythia koreana* | 79 | −23 |
| *Rhododendron mucronulatum* | 76 | −12 |
| *Prunus yedoensis* | 84 | −21 |
| *Prunus persica* | 88 | −11 |
| *Pyrus serotina* | 92 | −10 |

Earlier flowering species showed a bigger FFD shift than the later flowering species except for the Korean rosebay and cherry tree (Table 3). The advance of the FFDs of the early-season species in response to the temperature increase and earlier spring has been well documented and the results

of this study agree with those of the previous studies [31,32,40]. However, there is no evidence that the earlier flowering species are more sensitive to climate, and the regression analysis of the climatic factors (temperature and precipitation) and FFDs in this study showed the same result (Table 4). The determination coefficient ($R^2$) had no relation with FFD order. This suggests that different cues are likely responsible for initiating flowering. Nevertheless, it seems that the temporal asynchrony among the plant species will become worse with climate change as earlier flowering species will bloom more quickly and later flowering species will bloom less quickly. Moreover, a regression analysis of the FFDs of each plant species of the 72 observation sites with respect to the year was conducted to examine the variations in the phenological trend of each species (Figure 3). The slope of the regression (all showed negative slopes) and regional deviations showed that the FFDs of all investigated plant species advanced during 1920–2019. Each species has its own regional deviation value, and the FFD of the plum tree showed the greatest regional deviation, following the biggest FFD shift.

**Table 4.** Regression analysis results of phenology versus meteorological factors (confidence level of 99%).

| Species | Temperature | | Precipitation | |
|---|---|---|---|---|
| | Slope | $R^2$ | Slope | $R^2$ |
| *Prunus mume* | −6.37 | 0.60 | −0.27 | 0.12 |
| *Forsythia koreana* | −2.83 | 0.59 | −0.11 | 0.09 |
| *Rhododendron mucronulatum* | −2.81 | 0.50 | −0.10 | 0.06 |
| *Prunus yedoensis* | −2.73 | 0.66 | −0.10 | 0.08 |
| *Prunus persica* | −3.22 | 0.61 | −0.14 | 0.12 |
| *Pyrus serotina* | −2.45 | 0.45 | −0.08 | 0.05 |

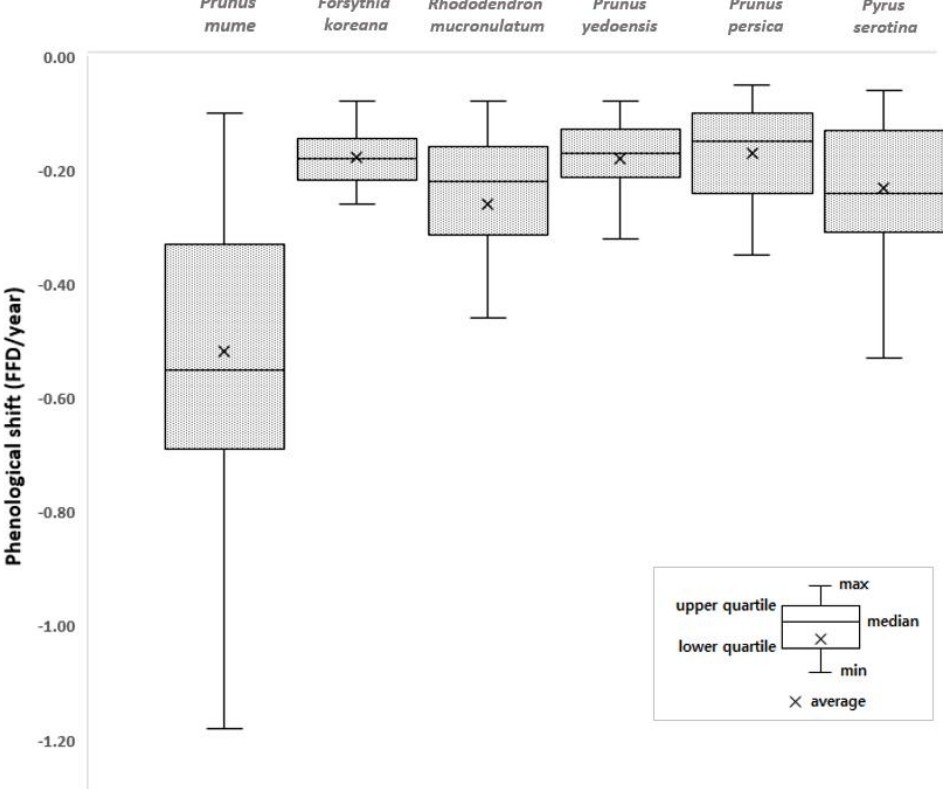

**Figure 3.** Shifts in flowering phenology of individual species during 1920–2019 in Korea. The regression of the phenological shift over time was investigated with $p < 0.05$; the height of each box represents the regional deviation.

### 3.2. Relationship between Plant Phenological Shift and Climatic Factors

The regression analysis of the climatic factors (temperature and precipitation) and FFDs showed that the FFDs of all investigated plant species occur earlier with increasing temperature and precipitation (Table 4). Our regression model suggests an advance in FFD of 2.4~3.3 days/°C increase in JFMT except for the plum tree. This value is within the range of other studies such as 1.5 days/°C in northwest America [39], 3.4 days/°C in northeast America [41], and 4 days/°C in England [11]. However, the plum tree, which is the earliest blooming species in this study, exhibited an exceptionally high magnitude of regression slope for the temperature (6.37 days/°C). The regression slopes for temperature in the species ranged from −6.37 to −2.45, and those of precipitation ranged from −0.27 to −0.08 for the species. The regression analysis comparison between precipitation and FFD has rarely been examined. However, there have been many reports that claim temperature effect on plant phenology surpassed effects of precipitation in Korea [22,33,42], and other countries [10,39]. The value of $R^2$ of our regression models also confirms that the temperature has a stronger effect on the FFDs than precipitation. In addition, according to previous study reports, precipitation may affect plant phenology long after the last rainy day, which makes it difficult to forecast the effect of precipitation on plant phenology [10,14]. Thus, in this study, the future FFDs were predicted based on only the temperature.

### 3.3. Prediction of FFDs in 2100

The FFDs in 2100 of all investigated species were calculated based on detailed climate change scenarios in South Korea (Figure 4). The difference in the FFDs between RCP 2.6 (the most positive scenario) and RCP 8.5 (the most negative scenario) was 12–22 days, and the difference for the plum tree was the biggest (21.83 days). In addition, the differences between RCP 2.6 and RCP 4.5 and between RCP 6.0 and RCP 8.0 were 1–3 days, while the difference between RCP 2.6 and RCP 6.0 was greater than 10 days for all species. Thus, the effectiveness of greenhouse gas policies will determine the degree of plant phenological shift in the future.

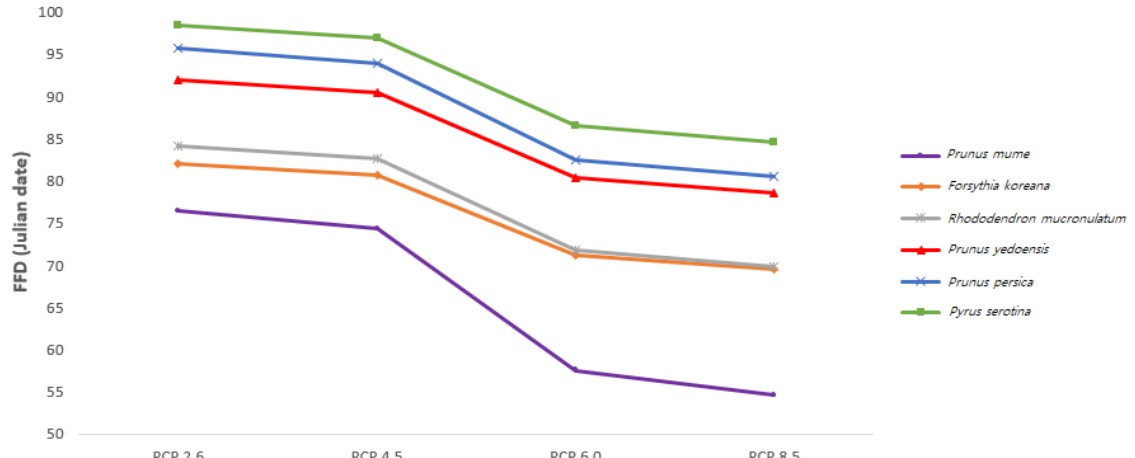

**Figure 4.** Predicted mean FFDs in 2100 in Korea based on representative concentration pathway (RCP) scenarios according to the Intergovernmental Panel on Climate Change (IPCC).

Based on a comparison of the FFDs of 2019 with those of 2100 in the most extreme scenario (RCP 8.5; Figure 5), the national averaged FFDs of each plant species are expected to be approximately 10–15 days earlier. The magnitude of these shifts varies spatially. The FFD of the Korean forsythia is 79–90 Julian days in 2019; it will advance to 54–86 Julian days in 2100. The FFD of the plum tree will advance from 37–96 Julian days in 2019 to (−3.3)–96 Julian days in 2100. More specifically, the plum tree will probably blossom in December on Jeju Island, which is the southernmost island in Korea. The FFD of the pear trees across the country will advance from 84–10 Julian days in 2019 to 55–100 Julian days in 2100. The cherry tree's FFD is expected to advance from 80–106 (2019) to 61–98 Julian days in the

country in 2100. Moreover, the peach tree's FFD will advance from 79–106 (2019) to 53–97 Julian days across the country in 2100. The FFD of the Korean rosebay is expected to advance from 72–98 (2019) to 51–89 Julian days in 2100. The spatial variations in the FFDs change with the latitude. The FFDs of each species in Seoul (at a higher latitude) will advance by approximately 10 days, while the FFD in Busan (at a lower latitude) will advance approximately 20 days or even 40 days for some species. Thus, the shift in plant phenology is likely to proceed more rapidly at lower latitudes. As the FFDs at lower latitudes are brought forward, it is assumed that the spatial variation in plant phenology will become worse; this will increase the asynchrony of plant phenology temporally and spatially.

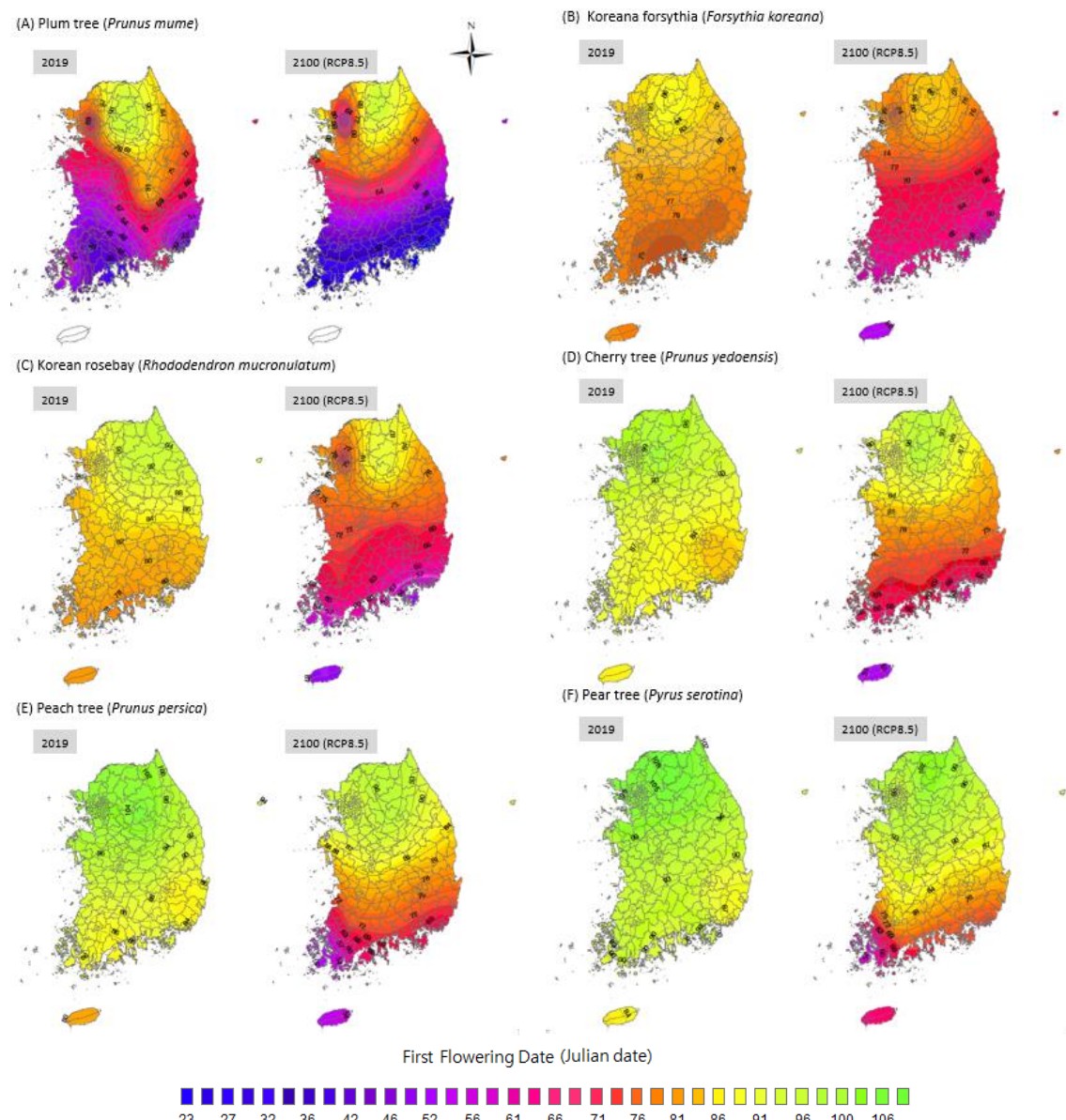

**Figure 5.** The FFDs of 2019 and predicted FFD of 2100 based on RCP8.5 scenario in Korea. Plant species names were labelled in (**A–F**).

However, winter chilling is also an important issue for plants. It is needed to sensitize plants to spring onset. According to previous research, the cold weather has decreased more rapidly than spring onset [43–45]. If winter is not chilly enough, more spring warming time is required for FFD [46], and this could slow the FFD and reduce reproductive success [47]. In addition, although warmer temperature could activate biochemical processes and lead to early FFDs as this paper ascertains,

high temperature could be destructive to plant species because this thermal adaptation is supposed to be concave up [48,49]. Plant species have thermal optima ranges and an increase in the temperature will cause these optima to fall [49]. The effect of winter chilling and the thermal optima ranges were not included in the analysis in this paper and this would be a limitation of this study. Thus, even though we predicted FFDs in 2100, we are not sure that the plants species would survive and actually bloom at that time. Comprehensive information about plants' life is needed to reliably quantify the plant phenological responses to climate change.

## 4. Conclusions

By using long-term phenological data from 72 observation sites in Korea, the phenological shift of plants in spring in the past was investigated, and predictions for 2100 were provided. The analyses show that the plant species have advanced their FFDs during the investigated 100 years. Each species showed a different shift rate but earlier flowering species showed greater shifts. The earliest flowering species (the plum tree) has advanced its FFD most; in addition, the shift exhibits the greatest regional variations. The temperature turned out to be the most relevant climatic factor for the plant phenology; it surpasses the effects of precipitation. These results suggest that climate change may aggravate the temporal asynchrony within the plant community.

The FFDs of each plant species in 2100 were predicted based on the relationship between temperature and the FFDs of the already investigate period. The FFDs differed by an average of 15 days depending on the RCP scenarios (RCP 2.0/4.5/6.5/8.5), and the plum tree, which has the earliest FFD, showed the largest deviation (approximately 22 days) for all scenarios. Furthermore, the RCP 4.5 and RCP 6.5 showed the biggest difference in FFDs. Thus, the shift of plant phenology will depend on how much greenhouse gas we will reduce. For the RCP 8.5 (the most extreme scenario), the nationwide averaged FFDs of each plant species is expected to be approximately 10 to 15 days earlier in 2100 than in 2019. Moreover, the shift rates of the FFDs of all investigated plant species are faster at lower latitudes. Consequently, the spatial asynchrony among the plant phenology across the country will become worse with climate change.

Climate change alters the environmental conditions (e.g., temperature and moisture) under which plant species live, and affects the ecosystem as a whole, thereby changing the interactions between species in a chain. In other words, environmental and phenological changes caused by climate change affect the habitat shift and ecological decoupling within communities, thereby determining whether or not the species will adjust and survive. The results of this study demonstrate that climate change advances the FFDs of spring-season plants, and intensifies the temporal and spatial asynchrony even within the same species. The ecological imbalance caused by climate change is not limited to phenological changes. Therefore, further studies with more diverse perspectives are required for multidimensional solutions.

**Author Contributions:** Conceptualization, writing and editing—original draft preparation and supervision, funding acquisition, S.D.L.; data analysis, H.K.L. and S.J.L.; validation and literature review, M.K.K. All authors have read and agreed to the published version of the manuscript.

**Funding:** This research was funded by Development of ICT-based Environmental Impact Assessment Technology (2020), NRF-2017R1D1A1B03029300 and Environmental Science & Technology Center (SEST), KOREA (2020).

**Conflicts of Interest:** The authors declare no conflict of interest.

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
