# Peer review of "Prediction of Plant Phenological Shift under Climate Change in South Korea"

_sustainability, doi:10.3390/su12219276_

Round 1

Reviewer 1 Report

Please elaborate introduction.

Conclusion should be in concise form.

Author Response

We updated the introduction, methods and discussion with addition of recent articles and publication in the literature.

Thank you

Reviewer 2 Report

In general the manuscript is well prepared, presenting interesting predictions on influance of climate change on plant growth. However, there are some minor issue that needs to be explained in more details. These are:

More information on the plant species monitoring is required. Please provide detail on the monitoring approach, data collection and methodology. 

Considering only temperature and precipitation as only factors for plant growth is very much debatable. Please give the reason why other factures were not concerned in the study.

Statistical analyses are not sufficiently covered to come up with such conclusions. Please provide more details on analyses especially in section 3.2.

Modelling process for 2100 estimation needs further explanation. There is not sufficient information to clearly understand the process of growth prediction.

Section 3.3 needs more explanation on the modelling approach and process.

The reference list needs updating, most of publications are not really recent

Fig.2 needs quality improvement. The labels are hard to read, the x axis is missing the title.

The order of presentation should be changed. The authors refer to tab.3 first and then to fig.3, please move the table and the fig.

Fig. 3 not sure what Tab. 0 refers to, is it the detail of the box?

The discussion is insufficient. More critical review of the study is needed, mainly due to lack of other parameters influencing plants’ growth that weren't included in the present study.

Author Response

Please find the reply to reviewer #2

Reviewer 3 Report

The manuscript investigates the effects of climate change on the First Flowering Date (FFD) of six plant species in South Korea, using phenology and climatic long timeseries of the previous 100 years and also projecting future changes until 2100, based on specific climate change RCP scenarios.

The FFD and temperature timeseries used in the paper, are quite extended and the phenological data series, probably continuously recorded in 72 locations in S. Korea.

Considering the long term data set, the relative high (6) number of plant species, I would definitely recommend the publication of the paper. However, the authors could have a better use, analysis and presentation of the data, concidering also the previous work (by one of the authors), which has been published in the same journal (see: Sustainablility 2017, 9, 2203 doi:10.3390/su9122203), though with relative smaller dataset (35 years from 1973 to 2008).

Unfortunately, I believe that the manuscript should be extensively revised and resubmitted. In my opinion it does not meet the high quality criteria to be published in the Journal Sustainability.

Some suggestions for improvements are presented below:

  1. The introduction section should be much extended, providing the most sound findings of similar works in Asia and worldwide. Unnecessary tables, like Table 1, could be removed.
  2. At the “Materials and Methods” section, the authors can give more information concerning the study sites, the data collection methods and mainly the uncertainties conserning the quality of the data and the data gaps. Also check the link you provide, in order the reader to have a direct access to the phenological data.

Also, explain why average temperature and precipitation of J, F, M, were considered as most influential. Give a justification based in other scientific works.

You can also give some characteristics of the climatic stations used in the paper i.e. altitude, average temperature, precipitation, climatic type e.t.c. Similarly, a table with information about the sites where the phenological data were recorded could also be added.

The title “Plant phenological model” of sub-section 2.3, does not describe the content of the text following.

  1. In the “Results and Discussion” section, I would anticipate to see the changes of phenology per species maybe in graffs like fig. 2.

The changes mentioned in lines 94-95 cannot be extracted from Fig. 2. A changing rate of +0.01oC/y would result to an average change of +1oC for the 100 year period (1920-2019) and not +2.6oC. Similarly, a changing rate of -0.03 mm/year in precipitation, results to about -3mm (not -18mm) average precipitation change for the same 100 year period.

Also, it is recommended to perform an analysis to show if these trends are significant. The same could be done for the FFD per species.

  1. The advancement of 10-59 days in FFD during the last 100 years, must be well documented. Such advancements (of 59 days i.e. about two months) for a specific species (not other clones or hybrids) are quite high. Similar numbers, should be also presented by reviewing other works either in Asia or worldwide. Also an extensive discussion could be added, since in a previous work by one of the authors, the advancement during the period 1973-2008 is mentioned to be between 3 to 18.6 days for the same species.

I would expect that an advancement of such magnitude in some species, would be destructive for the species, since they could not collect the necessary cool time (GDD) in order to “wake up” from the winter dormancy. Thus other temperature related parameters (cumulative cool time or minimum-maximum temperatures or maybe winter temperatures), should also be analyzed or at least discussed.

  1. The slopes mentioned in lines 130-131 an in Table 4, probably are in units days per oC or per mm. the authors mention that the greatest slopes of temperature compared to the respective of precipitation, indicate that “plant phenology is more sensitive to temperature than precipitation”. I can understand that a change of 1oC in temperature would have a more sound effect on FFD compared to a change of 1 mm in precipitation, but this cannot lead to the above-mentioned conclusion. Also according to the p values presented in Table 2, both temperature and precipitation correlations are indicated as significant. The authors should add a discussion about these issues.

Additionally, in lines 132-135, is written that “the reason why the regression results of precipitation and FFD showed a negative slope despite the decrease in precipitation during research period is thought to be due to the lack of significant decrease in the average precipitation and the increase in precipitation at some observation sites”. I cannot understand the meaning of this sentence. The slopes presented in Fig. 4 are negative i.e. as precipitation increases, the FFD decreases and the opposite. This is the relationship between the two parameters and has nothing to do, with the year to year precipitation changes mentioned above. Also, if the year to year precipitation changes are significant, why was precipitation considered as most influential factor (line 77)?

  1. In my opinion, the paper should only include the findings of the period 1920-2019. The addition of future projection is quite interesting but should be extensively presented, maybe in a new paper. However, if presented in the same paper you can add the current FFD values (probably average or per decade) in Fig. 4. It would be very useful to present the year to year or decadal changes of FFD in this graph and the predicted values for the different RCP scenarios.
  2. Extend the “Results and Discussion” section by adding a Discussion. Compare your results with other studies (same or other species) for the same or other regions.
  3. Avoid to include references in your conclusions. These statements can be added to the “Introduction” or “Results and Discussion” sections.
  4. Please consider and review the methodology, presentation and justification of results from your previous paper, which is well written and documented.

Author Response

Please find the reply to reviwer #3

Round 2

Reviewer 1 Report

Minor revision required.

Author Response

We revised the MS and uploaded. Thank you for reviewing the MS.

Reviewer 3 Report

Considering the revised manuscript entitled “Prediction of Plant Phenological Shift under Climate Change in South Korea” and the authors’ response to my comments, I believe that the manuscript has significantly improved. The text was extended by about 5 pages, but important information concerning the justification and the discussion of results were added, which is meaningful. The introduction and the discussion sections were enriched and 18 more references were added. The materials and methods section was significantly improved and important information was added.

However, some minor corrections could further enhance the quality of the manuscript.

Suggestions

In line 57, add the reference for the IPCC report

In lines 69-70, some spaces between words are missing.

In line 90, define the precise percentage of the “missing values”

In Fig. 2, the changing rates of temperature and precipitation were deleted at the second version of your paper. I believe that these trends (slopes) and the respective R2 values are meaningful and should be presented in all graphs of Fig. 2 (i.e. temperature, precipitation, FFDs), along with the p values.

In Table 1, the information included is very useful (especially the altitudes of the sites). In a future work a correlation of FFD with the altitudes would be extremely interesting. However, the geographical data of Table 1 are also presented in Fig. 1. I recommend to reduce the size of Fig. 1 by using dots or numbers to indicate the sites positions.

The two sub-graphs of Fig. 5 can be presented in one page be placing one next to the other.

A moderate English editing is required.

Author Response

Reviewer #3 comments

======================

Suggestions

In line 57, add the reference for the IPCC report

  • We added the reference: IPCC[50].

In lines 69-70, some spaces between words are missing.

  • Line spacing was corrected

In line 90, define the precise percentage of the “missing values”

  • Missing values (data neither recorded nor monitored)

In Fig. 2, the changing rates of temperature and precipitation were deleted at the second version of your paper. I believe that these trends (slopes) and the respective R2 values are meaningful and should be presented in all graphs of Fig. 2 (i.e. temperature, precipitation, FFDs), along with the p values.

  • We added ‘(A) Temperature JFMT = 0.01 C/year +1.57, (B) Precipitation JFMP=-0.03 mm/year +46.9 during 1920-2019 in Korea’ in Fig 2.

In Table 1, the information included is very useful (especially the altitudes of the sites). In a future work a correlation of FFD with the altitudes would be extremely interesting. However, the geographical data of Table 1 are also presented in Fig. 1. I recommend to reduce the size of Fig. 1 by using dots or numbers to indicate the sites positions.

  • Agreed and correct in the final version before publication

The two sub-graphs of Fig. 5 can be presented in one page be placing one next to the other.

  • In the editing those will be in one page

A moderate English editing is required.

  • Proof reading was done and additionally will be processed
